# System Design of Optimal Pig Shipment Schedule through Prediction Model

Jin-Wook Jang [1,*], Jong-Hee Lee [2], Gi-Pou Nam [1] and Sung-Ho Lee [3]

1 Department of Cooperative Digital Management, Agricultural Cooperative University, Goyang-si 10292, Gyeonggi-do, Republic of Korea; nkp17178@nonghyup.ac.kr
2 Department of Digital Agricultural Promotion, Agricultural Cooperative University, Goyang-si 10292, Gyeonggi-do, Republic of Korea; jhlee78@nonghyup.ac.kr
3 Hohyun F&C, Suwon-si 16432, Gyeonggi-do, Republic of Korea; dulee211@naver.com
* Correspondence: jjw@nonghyup.ac.kr

**Abstract:** We propose an optimal system for determining the shipping schedule for pigs using a predictive model using machine learning based on big data. This system receives photographic and weight measurement information for each pig from a camera and a weighing machine installed in a pig pen for raising pigs corresponding to a predetermined fattening period. Then, the photographic information of each of these pigs is applied to a predictive model machine-learned in advance to determine whether or not there are candidate pigs for determining the presence or absence of abdominal fat-forming pigs. And if there is a candidate pig, it is determined using a machine-learning model for predicting whether the candidate pig is an abdominal fat-forming pig by analyzing the pattern of weight increase of the abdominal fat-forming pig and changes in weight of a candidate. If the candidate pig is an abdominal fat-forming pig, the timing of shipping is determined by predicting when the weight of the candidate pigs, specifically the abdominal fat-forming pigs, will reach a predetermined minimum shipping weight. This prediction is made using a machine-learning model that considers the weight gain trend pattern of abdominal fat-forming pigs and tracks changes in the weight of the candidate pig. A machine-learning model is used to predict the timing of weight gain in candidate pigs, specifically those that develop abdominal fat, in order to determine the optimal shipping time. By analyzing the weight gain patterns of abdominal fat-forming pigs and monitoring the weight changes in the candidate pig, the model can predict when the candidate pig will reach the minimum weight required for shipping. In this paper, we would like to present a point of view based on the body type and weight of pigs corresponding to the fattening period through this system, whether intramuscular fat has adhered or abdominal fat is excessively formed by the fed feed and appropriate shipment as the fattening status of pigs.

**Keywords:** machine learning; shipping; prediction model; shipping timing; convolutional neural network

## 1. Introduction

### 1.1. The Breeding Process on a Pig Farm

In the pig farming industry, pigs for meat production are born after a 16-week gestational period. After being separated from their mothers, they are raised for approximately 180 days before being shipped. During the lactation period, which lasts up to the fourth week after birth, piglets receive breast milk from their mother pigs. From the fourth to the eighth week, they are weaned off the mother and transitioned to compound feed during the piglet period. Following this, they enter a growing period until the 22nd week, where they are provided with a high-protein feed to develop muscles. In the 26th week, the pigs enter the fattening period, consuming a high-fiber diet to enhance intramuscular fat and produce high-quality meat [1]. The quality of meat and grading of shipping pigs can vary depending on feeding and rearing practices during the breeding period. Management

and breeding techniques employed during the 22nd to 26th week, known as the fattening period, are particularly crucial for producing premium, high-quality meat.

In terms of body composition of these pigs, fat accumulation in fattening pigs consists of subcutaneous fat, root fat, intramuscular fat, and abdominal fat. Based on 100 kg of live weight, the fat accumulation is estimated in Table 1. Among these, the fattened pigs with excessively formed abdominal fat are rated very low in conductor grade and meat quality score, and accordingly, the price shipped is also rated very low. However, even though some pigs accumulated severe abdominal fat and their commercial value decreased during the fattening period, it was impossible to determine which pigs among the pigs shipped had a significant accumulation of abdominal fat. Figure 1 means the degree of fat by pig part. The characteristics of each part are shown in Table 1.

(a)                                                    (b)

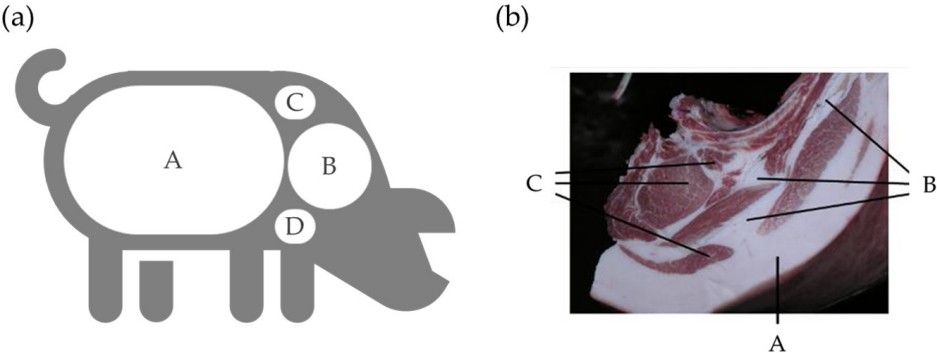

**Figure 1.** Illustration (**a**) and real sample (**b**) of fat accumulation in fattening pigs in terms of body composition.

**Table 1.** Fat accumulation in fattening pigs in terms of body composition.

| Part | Type | Rating (%) |
| --- | --- | --- |
| A | Subcutaneous Fat | 70~75 |
| B | Intermuscular Fat | 20 |
| C | Intramuscular Fat | 1~2 |
| D | Abdominal Fat | 5 |

Most of the pigs are shipped after the designated fattening period has passed. Meanwhile, the pigs that accumulate abdominal fat when ingesting feed causes the breeding cost to increase since it is not possible to secure income shipped as much as feed costs in the fattening period [2].

Shipping weight of fattening hogs is the most profitable economic factor on the producer side, and one of the main factors in determining conductor grade or quality on the consumer or slaughterhouse side. In reality, the shipping weight is relatively variably adjusted according to the conductor quality, or the shipping weight is set within a certain limit, and the genetic characteristics and specification methods of the fattened pigs are adjusted according to the conductor quality or grade criteria. There is a way to select or adjust the Up to a certain live weight, the higher the shipping weight, the lower the production cost of pork per unit weight. In particular, changes in production cost and body fat ratio of pork due to changes in live weight act as major factors in determining shipping weight [3,4].

For these factors, the recent domestic pig farming industry will be able to predict shipment schedules more accurately by leveraging smart pig pens based on the Internet of Things. For the efficient management of a smart pig pen, the environment inside the pig gourd and the biological information of each individual are collected and analyzed [5]. Studies are underway to predict growth in pigs based on came [6,7]. In particular, it is connected to machine learning analysis by utilizing big data collected from various sensors within a smart pig pen.

*1.2. The Prediction Models*

The prediction model can be machine-learned using continuously photographed photo information for pigs corresponding to the fattening period, ultrasound data measuring fat distribution in the body at each point in time, and continuously measured weight information as learning data [5].

In addition, the received photographic information and weight data for a random pig in the pig pen based on the change in body shape information value and weight of the corresponding pig can be used to implement an artificial intelligence-based module to determine whether it is an abdominal fat-forming pig.

Furthermore, the prediction model can generate prediction information on the point when the weight of the abdominal fat-forming pig existing in the pig pen reaches a pre-determined minimum shipping weight by machine learning using learning data on body shape changes and weight changes of the abdominal fat-forming pig.

In addition to learning by using the existing accumulated photo information, ultrasonic data, weight information, and other breeding history data of the pigs corresponding to the fattening period at the livestock farm can be used as learning data. It also becomes possible to learn breeding history data generated from other livestock farms or collected on the internet as learning data [6–8].

Prediction models use generative models such as the few-shot running method, which can be learned with only a small amount of training data, and general adversarial networks (GANs), which are generated with a small amount of training data. Using this model, learning to use a method of solving a shortage of learning data by increasing similar data generated at another place with the same environment can be implemented.

The predictive model may be generated as one or more deep learning-based models, such as a fully convolutional neural network, a convolutional neural network, a recurrent neural network, a restricted Boltzmann machine, a deep belief neural network, and the like. It is also possible to use machine-learning technology other than deep-learning technology or to generate a hybrid model that combines deep-learning technology and machine-learning technology [9,10].

A method of learning a prediction model can also be classified into supervised learning, unsupervised learning, reinforcement learning, and the like. The reference models for this study are shown in Table 2 below, and each model uses a classification and regression tree based on the gradient descent method.

**Table 2.** Reference model.

| Model | | Method |
|---|---|---|
| GBC | Gradient Boosting Classifier | Gradient descent |
| XGBoost | Extreme Gradient Boosting | Classification, regression tree |
| LightGBM | Light Gradient Boost Machine | Level-wise |
| CatBoost | - | Level-wise, ordered boosting |
| KNN | K Nearest Neighbor | Decision Tree, data distance |

The system proposed in this study, "System Design of optimal pig shipment schedule through pre-diction model", can provide the following services. Based on the body type and weight of the pig corresponding to the fattening period, it is possible to estimate whether the intramuscular fat is sticky or the abdominal fat is excessively formed by the feed and suggest an appropriate shipping time according to the fattening situation of the pig [9].

In addition, in the case of a fattening pig with excessively formed abdominal fat, it is possible to ship it earlier than an ordinary fattening pig so that feed costs may be reduced, and good meat quality may be endured, thereby improving the income of livestock farmers.

## 2. Method

In this study, a system with the following configuration was designed to execute step-by-step determining methods and the shipping timing for pigs.

### 2.1. System Configuration

The system proposed in this paper is designed assuming the installation of hardware such as a camera for object recognition, a weight measurement sensor, and a supply quantity measurement sensor. This system includes a device for determining the shipping timing for pigs and object recognition units such as a camera unit and a weight measurement unit, which are both installed in a pig pen where pigs corresponding to a fattening period are raised. The device for determining the shipping timing for pigs includes a receiver, a predictor, a report generator, storage and a controller, a web crawler, and an analyzer [5].

The receiving unit consists of an individual recognition unit, a camera unit, and a weight measurement unit installed in pig pens where pigs correspond to the fattening period. Moreover, those are connected by a wired or wireless communication method to receive individual recognition information, photo information and weight measurement information, respectively [6].

Figure 2 is a schematic configuration diagram of the system for determining the shipping timing for pigs to be designed in this research.

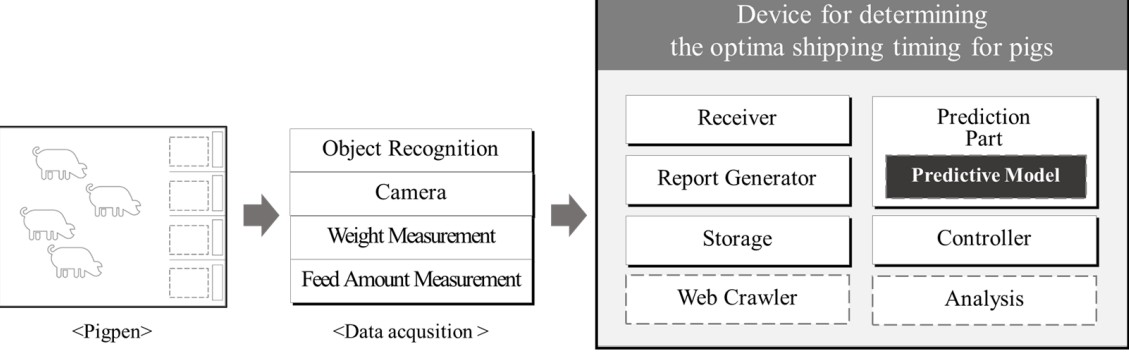

**Figure 2.** System for determining the optima shipping schedule for pigs.

The prediction unit applies the breeding history data for each pig, i.e., photographic information, weight measurement information, etc., to a prediction model machine-learned in advance and determines whether the pig is normal or an abdominal fat-forming pig. This can be used to generate predictive information for the optimal timing of shipping for abdominal fat-forming pigs.

The breeding history data of each pig can be generated by the camera unit and the weight measurement unit of the pig pen where the pig belongs to the breeding period, received through the receiving unit, and could be stored in the storage unit [10].

As shown in Figure 3, the body type information value for determining whether abdominal fat is formed can be calculated using the height (A) and body height (B) of the pig interpreted from the side photo information of the pig and the width (C) of the body interpreted from the top photo information. Also, it may be generated in the prediction model or determined to generate by the prediction unit [11,12].

### 2.2. The Environment of a Pig's Pens

In order to collect basic information about the system, several camera units and weight measurement units are installed in the pig pens.

Several feeding units are installed at a height where pigs can take in feed in a standing position. And it is possible to respond to the feeding unit so that one pig can enter and then take in feed correctly when a partition is installed. A camera unit is installed to photograph

a pig that enters a feeding table divided into partitions in a pig pen and ingests feed that is supplied in a standing position.

(a) 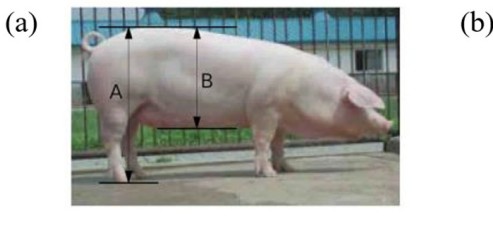  (b) 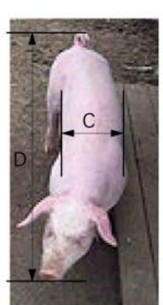

**Figure 3.** Measurement part of the pig's body shape information valve used in the calculation ((**a**) A: height, B: body height; (**b**) C: width, D: length of the pig).

A number of camera units is installed to generate top-side photographic information of a pig in a standing position as shown in Figure 3. Also, a number of second camera units is installed to generate side photographic information of a pig in a standing position from the side. In addition, the Radio Frequency Identification reader for recognizing the RFID tag attached to the pig was installed for the object recognition part [13].

In this way, in the process of a pig entering the pig pen and taking in feed in a standing position, individual recognition information of a pig that takes in feed can be generated. Moreover, this pig's photo and weight measurement information can be managed by corresponding to this object recognition information.

### 2.3. Normal Pigs and Abdominal Fat-Forming Pigs

Normal pigs, as intended, refer to pigs with muscle and intramuscular fat formed from the 22nd to 26th week after birth, and abdominal fat-forming pigs refer to pigs mainly formed with abdominal fat during the fattening period. Of course, the division of normal pigs and abdominal fat-forming pigs is not distinguished by the presence or absence of abdominal fat formation. It is because even normal pigs may produce an appropriate amount of abdominal fat during the breeding process up to the time of shipment. It should be understood whether abdominal fat is produced at a level that is acceptable for the pig to be classified as normal pig, so that the marketability which pertains to whether the pig is a conductor grade and meat quality grade, is not affected.

Figure 4 shows the difference in weight gain between normal and abdominal fat-forming pigs during the fattening period [14–16].

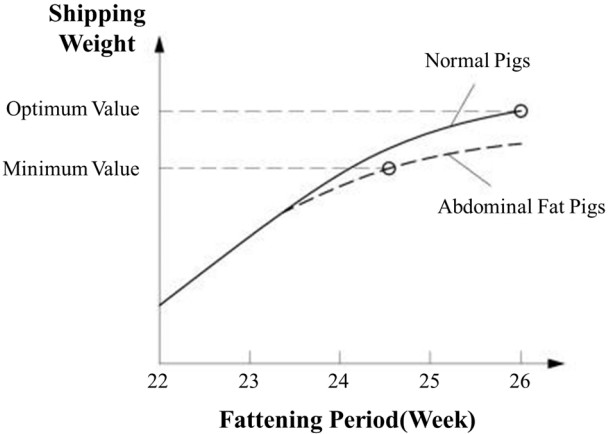

**Figure 4.** The difference in weight increases between normal pigs and abdominal fat-forming pigs during the fattening period.

### 2.4. The Process for Determining the Shipping Timing for Pigs

The device for determining the shipping timing for pigs is a machine-learned prediction model using breeding history data designated as learning data.

At this time, the breeding history data corresponds to photographic information, ultrasonic data, and weight measurement information accumulated in the existing breeding process for pigs corresponding to the fattening period in the livestock farm. The learning data may also include breeding history data generated by other livestock farmers, breeding history data that can be collected through the Internet, and so on [17].

Since the prediction model is machine-learned using learning data, it is possible to distinguish between normal pigs and abdominal fat-forming pigs using photographic information and weight measurement information of pigs raised in the pig pen during the fattening period. It can also be used to determine the optimal shipping time for pigs based on the result of the determination.

The device for determining the shipping timing for pigs receives each pig's photographic information and weight measurement information corresponding to the fattening period, which is breeding in the pig pen. In addition, an object recognition unit, a camera unit, and a weight measurement unit can be provided in the pig pen to receive each pig's photographic information and weighing information.

The device for determining the shipping timing for pigs uses a prediction model machine-learned in advance to interpret the photographic information of each pig. Then, whether or not the body shape information value thus analyzed exceeds the target threshold value is determined.

The body shape information value of pig farming is analyzed and calculated as follows.

1. Analyze the height value and torso height value of the pig using the video analysis technique specified in advance from the information of the top view photographed by the camera unit.
2. Interpret the torso width value of the pig from the lateral photographic information.
3. An interpreted torso height and width value through 2 can be calculated by dividing the estimated torso circumference by the height value of the pig.

As a result of the determination, if no pig has a body type information value equal to or greater than the reference threshold value, it is determined that all the pigs are normal, and the step proceeds again. If a pig has a body type information value equal to or greater than the threshold due to the determining process for the shipping timing for pigs and the device for determining the shipping timing for pigs selects the corresponding pig as a candidate pig, a prediction model that has been machine-learned in advance can be used to determine whether or not the weight change transition of the candidate pigs measured so far, matches the weight gain transition pattern of abdominal fat-forming pigs derived by machine learning in advance [18].

If it does not match the weight gain change pattern of abdominal fat-forming pigs, the candidate pig is determined to be a normal pig, and the process proceeds again [18].

However, suppose the weight change transition of the candidate pig is matched with the weight gain transition pattern of abdominal fat-forming pigs, the device for determining the shipping timing can be used to determine that the candidate pig is an abdominal fat-forming pig. Then, using a machine-learned predictive model in advance, predictive information regarding the point in time when the body weight of this pig increases to a pre-determined minimum shipping weight is generated based on the weight gain transition pattern [19,20].

Figure 5 shows the process of determining when pigs are shipped.

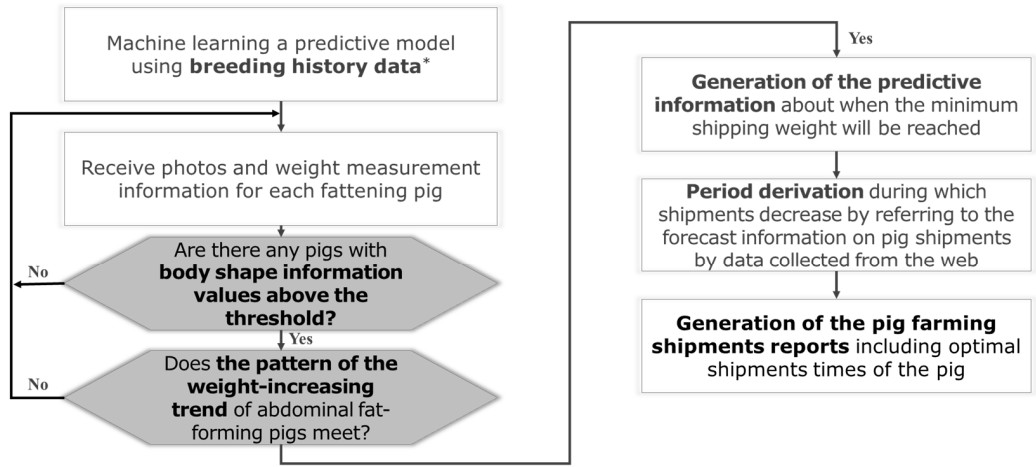

**: photo information/ultrasound data/weight measurement information and so on*

**Figure 5.** The process of determining when breeding pigs are shipped.

### 3. Model-Based Pig Release Timing Method Process Simulation

The model was selected and designed based on the feeding amount of the most interest to the farmers using the process for determining the timing of pig release presented in this study [8].

A total of 12,040 feeding data from pig farms were collected and pre-processed, and for missing data, the average value was applied, linear regression method was used to supplement. In order to verify the process of determining the shipping timing for pigs proposed in this paper, five models were implemented, and two models were selected. The accuracy of the model was evaluated using Equation (1), and it is possible to plan to improve the accuracy by using additional feeding data in the future.

$$Acuracy = \frac{1}{n} \sum_{i=1}^{n} I(y_i \neq f(x_i)) . \tag{1}$$

As shown in Table 3, the selected model consists of 8 models xgboost, lightgbm, et, rf, knn, dt, gbc, lr, and that are high-accuracy models. The models were compared and the model with the highest accuracy was selected.

**Table 3.** Model selection.

|  | Model | Accuracy | AUC | Recall | Prec. | F1 | Kappa | MCC |
|---|---|---|---|---|---|---|---|---|
| xgboost | Extreme Gradient Boosting | 0.9410 | 0.9958 | 0.9409 | 0.9413 | 0.9410 | 0.9311 | 0.9312 |
| lightgbm | Light Gradient Boosting Machine | 0.9401 | 0.9956 | 0.9401 | 0.9405 | 0.9402 | 0.9302 | 0.9302 |
| et | Extra Trees Classfier | 0.9380 | 0.9953 | 0.9381 | 0.9378 | 0.9370 | 0.9277 | 0.9280 |
| rf | Random Forest Classfier | 0.9352 | 0.9948 | 0.9352 | 0.9349 | 0.9348 | 0.9244 | 0.9244 |
| knn | K Neighbors Classfier | 0.8351 | 0.9618 | 0.8354 | 0.8355 | 0.8189 | 0.8077 | 0.8117 |
| dt | Decision Tree Classfier | 0.8089 | 0.8885 | 0.8088 | 0.8065 | 0.8072 | 0.7771 | 0.7772 |
| gbc | Gradient Boosting Classfier | 0.7867 | 0.9610 | 0.7865 | 0.7865 | 0.7860 | 0.7512 | 0.7514 |
| lr | Logistic Regression | 0.4208 | 0.7779 | 0.4208 | 0.4209 | 0.4195 | 0.3242 | 0.3245 |

In the case of the top five models, the blending accuracy improved considering that the amount of real data used for training was small for most important indicators as a result of basic tuning and blending. Accuracy improvement can be expected through the model selection and blending process. The blending accuracy and measurement results are shown in Table 4. As a result of blending, Accuracy 0.95 and F1 0.95 were obtained.

**Table 4.** Blending Accuracy.

| Fold | Accuracy | AUC | Recall | Prec. | F1 | Kappa | MCC |
|------|----------|-----|--------|-------|-----|-------|-----|
| 0 | 0.9484 | 0.9973 | 0.9484 | 0.9477 | 0.9477 | 0.9398 | 0.9399 |
| 1 | 0.9559 | 0.9981 | 0.9559 | 0.9559 | 0.9559 | 0.9485 | 0.9485 |
| 2 | 0.9506 | 0.9978 | 0.9506 | 0.9501 | 0.9501 | 0.9424 | 0.9425 |
| 3 | 0.9514 | 0.9971 | 0.9514 | 0.9512 | 0.9513 | 0.9433 | 0.9433 |
| 4 | 0.9540 | 0.9973 | 0.9541 | 0.9538 | 0.9538 | 0.9464 | 0.9464 |
| Mean | 0.9521 | 0.9975 | 0.9521 | 0.9517 | 0.9518 | 0.9441 | 0.9441 |
| Std | 0.0026 | 0.0004 | 0.0026 | 0.0028 | 0.0028 | 0.0031 | 0.0030 |

Weighted avg is a high number, and it can be seen that the accuracy is placed at a certain level and can be interpreted as being in a stable range. The accuracy is 792, the weighted average is 792. It can be determined that the accuracy of a certain average is maintained. As for precision, the value predicted by the model is in the confidence interval with an average of 0.61. Recall is 0.69, and the reliability of the model is high. The F1-score is the median value of precision and recall and is 0.65. Test results are shown in Table 5.

**Table 5.** Test results.

| Fold | Precision | Recall | F1-Score | Support |
|------|-----------|--------|----------|---------|
| 0 | 0.17 | 0.04 | 0.05 | 28 |
| 1 | 0.16 | 0.17 | 0.16 | 42 |
| 2 | 0.78 | 0.91 | 0.84 | 590 |
| 3 | 0.20 | 0.17 | 0.19 | 23 |
| 4 | 0.00 | 0.00 | 0.00 | 22 |
| 5 | 0.05 | 0.02 | 0.03 | 50 |
| 6 | 0.17 | 0.05 | 0.08 | 37 |
| accuracy | | | 0.69 | 792 |
| Macro avg | 0.22 | 0.19 | 0.19 | 792 |
| Weighted avg | 0.61 | 0.69 | 0.65 | 792 |

Feeding data by pig group should be further supplemented. Then, it is judged that a more accurate confidence interval can be obtained. for balanced data collection, verification of the data collection process and collection results of farms should be strengthened. It is thought that more accurate model learning will be possible if the quantity of data for each group is secured.

## 4. Discussion and Conclusions

According to the simulation results of this study, based on the body type and weight of the pig during the fattening period, it is adequate to estimate whether the intra-muscle is sticking, or abdominal fat is excessively formed by the feed. And it suggests an appropriate shipping time according to the fattening situation of the pig.

In addition, in the case of fattening pigs in which abdominal fat is excessively formed, shipping earlier than a normal fattening pig makes it possible to improve livestock farmers' income because it can reduce feed costs and ensure good meat quality. Moreover, it is also effective in improving the profits of livestock farmers by suggesting that they ship excessively formed abdominal fat at a time when the shipping price is expected to be high by collecting pig industry outlook information on the web.

The effects obtained in this study are not limited to the effects mentioned so far and other effects not mentioned may be understood by those skilled in the art this study belongs to from the following description and a process for determining the timing of breeding pigs was proposed. Through this, the efficiency of shipping time was improved.

A total of 12,040 feeding data from pig farms were collected and pre-processed, linear regression method was used to supplement. In order to verify the process of determining the shipping timing for pigs proposed in this paper, five models were implemented and

two models were selected. The accuracy of the model was evaluated. The value predicted by the model is in the confidence interval with an average of 0.61. Recall is 0.69, and the reliability of the model is high. The F1-score is the median value of precision and recall is 0.65.

Verification of the data collection process and collection results of farms should be strengthened. It is thought that more accurate model learning will be possible if the quantity of data for each group is secured.

Through the process of determining the method for shipping timing for pigs presented in this study, it is possible to predict the supply amount and contribute to presenting an appropriate timing according to the fattening situation. In this study, deep learning technology was applied to pig farms. This allows more quantitative management. It is hopeful that this will help improve the profit structure of pig farms.

**Author Contributions:** Data curation, J.-W.J. and S.-H.L.; Investigation, G.-P.N.; Methodology, J.-W.J. and J.-H.L.; Project administration, G.-P.N.; Software, J.-W.J. and S.-H.L.; Supervision, S.-H.L.; Writing—original draft, J.-W.J.; Writing—review & editing, J.-W.J. and J.-H.L. All authors have read and agreed to the published version of the manuscript.

**Funding:** This work was supported by Korea Institute of Planning and Evaluation for Technology in Food, Agriculture and Forestry (IPET), and Korea Smart Farm R&D Foundation (KosFarm) through Smart Farm Innovation Technology Development Program, funded by Ministry of Agriculture, Food and Rural Affairs (MAFRA) and Ministry of Science and ICT (MSIT), Rural Development Administration (RDA) (421029-04), and The National Research Foundation of Korea in 2023 (No. 2022S1A5A8049255).

**Institutional Review Board Statement:** Not applicable.

**Data Availability Statement:** Data is contained within the article.

**Conflicts of Interest:** The authors declare no conflict of interest.

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
