# Peer review of "System Design of Optimal Pig Shipment Schedule through Prediction Model"

_agriculture, doi:10.3390/agriculture13081520_

Round 1

Reviewer 1 Report

Dear Authors,

Overall the manuscript is well written and the goal is clear. There are a few places where the english grammar is strange, and could be re-written. I have a few suggestions to improve your work:

1) In the Introduction, the first 3 paragraphs on the lifespan could be combined into one paragraph.

2) Figure 2 does not seem relevant to this work. It is a generic CNN image. Best is to remove it as it has low impact, and/or replace it with some cartoons of all the models tested here.

3) Section 3 could be named as "Methods" rather than "Main issue" which sounds like a problem.

4) In Figure 4 caption, it is suggested to write what A, B, C actually mean. And you could put both pictures next of one another horizontally.

5) Section 3.3 on "Normal pigs and abdominal fat-forming pigs" is not clear in the explanation. The figure is clear. The text could be re-looked at and simplified to explain the definition of abdominal fat-forming pigs compared to normal pigs.

6) In Section 4, it is not clear what the 12,000 data are of? Are these data images of pigs or weights or something else? Figure 7 and 8 could be re-drawn in white background. The yellow highlights in Figure 7,8  makes it hard to read the numbers. The confusion matrix is blurred. It is suggested to add a paragraph on which model performed best and why compared to others, so you could help the readers understand the pros and cons of each model.

7) In the discussion section, this sentence can be removed as it is hard to understand :The effects obtained in this study are not limited to the effects mentioned so far and 271 other effects not mentioned may be understood by those skilled in the art this study be- 272 longs to from the following description.

8) In the Discussion section, you could mention that additional health monitoring tools could be accompanied with your machine learning model to ensure the pigs are healthy at the time of shipping. Wearable sensors and cameras for activity monitoring can be used with your model. One recent article on collecting pig activity data discusses this point - "Behavioral Monitoring Tool for Pig Farmers: Ear Tag Sensors, Machine Intelligence, and Technology Adoption Roadmap". Animals, 2021, 11, 2665. https://doi.org/10.3390/ani11092665  

There are few minor grammar mistakes. I suggest to read every section and reword any sentence that seems weird in english.

Author Response

Hello, we will send you the contents of the review.

1. 

Before:
In the pig farming industry, fattening pigs for producing meat are born after a gestation period of the 16th week. After the suckling period, these piglets are separated from their mother pigs, raised for up to about 180 days, and then shipped. Fattening pigs receive breast milk from the mother pig during the lactation period up to about the 4th week after birth. Then over the lactation period, they are separated from the mother pig until about the 8th week after birth and go through the piglet period which feeds a compound feed.
They are supplied with high-protein feed until the 22nd week of life after the piglet period undergoes an upbringing period in which muscles are generated and after this period. On the 26th week, these pigs undergo a fattening period in which they are fed a diet with a high fiber content so that the intramuscular fat sticks to produce high-quality meat.
The fattened pigs are shipped out and slaughtered when this fattening period has elapsed and is about the 26th week of life [2]. Therefore, conductor grades and meat quality scores in shipping pigs can be very variable, depending on what kind of feed they supplied or how they managed the rearing environment during the breeding period. Especially, to produce high-quality, premium meat, how pigs were bred and managed in the fattening period, which corresponds to about the 22nd to 26th week of life, is very important.

After:
In the pig farming industry, pigs for meat production are born after a 16-week gestational period. After being separated from their mothers, they are raised for approximately 180 days before being shipped. During the lactation period, which lasts up to the 4th week after birth, piglets receive breast milk from their mother pigs. From the 4th to the 8th week, they are weaned off the mother and transitioned to compound feed during the piglet period. Following this, they enter a growing period until the 22nd week, where they are provided with a high - protein feed to develop muscles. In the 26th week, the pigs enter the fattening period, consuming a high-fiber diet to enhance intramuscular fat and produce high-quality meat. The quality of meat and grading of shipping pigs can vary depending on feeding and rearing practices during the breeding period. Management and breeding techniques employed during the 22nd to 26th week, known as the fattening period, are particularly crucial for producing premium, high-quality meat.

2. image deleted

3. change to  Method

4. Figure 3 re-edited

  Figure 3. The process of calculating the body shape information value of a pig.  Measurement part of the pig’s body shape information valve of used in calculation (A : height, B: body height, c: width, D : length of the pig)

5. Of course, the distinction between normal pigs and pigs with abdominal adipogenesis does not depend on whether abdominal fat is not produced at all. In other words, normal pigs can produce an appropriate amount of abdominal fat during the breeding process until shipment, and whether normal pigs and abdominal fat forming pigs produce abdominal fat at a level that can affect marketability, such as carcass grade and meat quality grade.

6. 7. The table was created and filled in. 12000 data is raw data. This was the content I entered last time because there were about 12,000 data. The pros and cons of each model and comparison details were not included.

8. Regarding the technology roadmap, we are experimenting at the stage of reviewing the application to future equipment.

9. It will be supplemented through native speaker reviews

The revised thesis is attached. 
Thank you.

Reviewer 2 Report

This research uses machine learning models to predict whether the candidate pig is an abdominal
fat-forming pig. The topic is interesting and can make contributions to the field of applying
advanced machine learning models to agriculture.
However, I have the following concerns.
1. The title–“System Design on determining the optimal pig shipping schedule using a prediction
model which applied machine learning based on big data”–should be revised. “which” seldom
appears in the title. And in most cases, machine learning indicates big data.
2. The literature review is incomplete. Research gaps and research contributions should be
clearly summarized.
3. Table 2 looks weird. Why is GBC highlighted?
4. In the methodology part, “i” should be the subscript in line 246.
5. The results should be expressed in a table, not the screenshot of the experiment results.
6. As this study only applies the off-of-shelf machine learning models to address specific
practical problems, the authors are suggested to add analysis about how these machine
learning models fit the research problem and why it can bring benefit to apply these models.

I recommend a major revision.

The language should be organized in an academic and rigorous manner. For example, line 81, "the prediction model can be machine-learned". To my knowledge, no one would use "machine-learned" to express the machine learning model.

Author Response

Hello, we will send you the contents of the review.

  1. befor) System Design on determining the optimal pig shipping schedule using a prediction model which applied machine learning based on big data
    after) System Design of optimal pig shipment schedule through pre-diction
  2. Add related literature
  3. has been modified
  4. has been modified
  5. has been modified
  6. Please understand that performance has been improved through blended. "A bright number is a high number, and it can be seen that the accuracy is placed at a certain level and can be interpreted as being in a stable range. Accuracy is 792, weighted average is 792. It can be determined that the accuracy of a certain average is maintained. As for precision, the value predicted by the model is in the confidence interval with an average of 0.61. Recall is 0.69, and the reliability of the model is high. The F1-score is the median value of precision and recall, and is 0.65"
  7. It will be supplemented through native speaker reviews

The revised thesis is attached. 
Thank you.

Reviewer 3 Report

It seems that the authors try to predict the shipping timing of the pigs using the body size and weight information. Overall the paper is too brief and hard to understand, the result also does not seem to make sense.

(1) The logic is not clear due to the limited language quality. For example, in the abstract, line 23 - 27, it is one sentence that is very confusing

(2) There is no sufficient introduction about the background of the research

(3) Only 4 references have been included in the related work

(4)  The paper is too short, and both the major method and results are not well described. e.g. confusion matrix in figure 9 does not show a high classification accuracy

It is generally hard to understand

For example, in the abstract, line 23 - 27, it is one very long sentence that is very confusing

Author Response

Hello, we will send you the contents of the review.

1. 

befoer)

If the candidate pig is an abdominal fat-forming pig, an information of the shipping timing is generated and predicted by predicting the time when the weight of the candidate pigs which are abdominal fat-forming pigs will increase to a predetermined minimum shipping weight using a machine-learning model for prediction and referring to the gain trend pattern of the weight of an abdominal fat-forming pig and changes in weight of a candidate pig.

after)

If the candidate pig is an abdominal fat-forming pig, the timing of shipping is determined by predicting when the weight of the candidate pigs, specifically the abdominal fat-forming pigs, will reach a predetermined minimum shipping weight. This prediction is made using a machine-learning model that considers the weight gain trend pattern of abdominal fat-forming pigs and tracks changes in the weight of the candidate pig.

A machine-learning model is used to predict the timing of weight gain in candidate pigs, specifically those that develop abdominal fat, in order to determine the optimal shipping time. By analyzing the weight gain patterns of abdominal fat-forming pigs and monitoring the weight changes in the candidate pig, the model can predict when the candidate pig will reach the minimum weight required for shipping.

2. 

after) Shipping weight of fattening hogs is the most profitable economic factor on the producer side and one of the main factors in determining conductor grade or quality on the consumer or slaughterhouse side. In reality, the shipping weight is relatively variably adjusted according to the conductor quality, or the shipping weight is set within a certain limit, and the genetic characteristics and specification methods of the fattened pigs are adjusted according to the conductor quality or grade criteria. There is a way to select or adjust the Up to a certain live weight, the higher the shipping weight, the lower the production cost of pork per unit weight. In particular, changes in production cost and body fat ratio of pork due to changes in live weight act as major factors in determining shipping weight. [3-6]

   For these factors, the recent domestic pig farming industry will be able to predict shipment schedules more accurately by leveraging smart pigpens based on the Internet of Things. For the efficient management of a smart pigpen, the environment inside the pig gourd and the biological information of each individual are collected and analyzed. Studies are underway to predict growth in pigs based on came. [7-86] In particular, it is connected to machine learning analysis by utilizing big data collected from various sensors within a smart pigpen.

3. Add reference

4. . I gave an explanation "A bright number is a high number, and it can be seen that the accuracy is placed at a certain level and can be interpreted as being in a stable range. Accuracy is 792, weighted average is 792. It can be determined that the accuracy of a certain average is maintained. As for precision, the value predicted by the model is in the confidence interval with an average of 0.61. Recall is 0.69, and the reliability of the model is high. The F1-score is the median value of precision and recall, and is 0.65."

5. It will be supplemented through native speaker reviews

The revised thesis is attached. 
Thank you.

Round 2

Reviewer 2 Report

The revised version is improved. But there are still some typos. After revising them, I think the manuscript can be considered for publication.

1. Table 2. There is a typo in "Light Gradient Boost Machin"

2. In line 260 eq.1, the symbol "i" should be subscript.

N.A

Author Response

Review contents were well checked.

1. Table 2. There is a typo in "Light Gradient Boost Machin"
Machin->Machine typo fix
2. In line 260 eq.1, the symbol "i" should be subscript.
Edit with formula subscripts

Attached.
thank you

Reviewer 3 Report

The authors have added more details about the background of the research. However, the working principle of the major method proposed in the paper is still not clear.

line 254, the authors mentioned 12000 data from pig farms were collected. While it's not clear how the data are applied to the machine learning classification models

(1) what is the format of these data?

(2) what is the data distribution? e.g., how many pigs?

(3)How does linear regression work on the missing data? It is suggested to give some examples.

(4) line 260, the definition of accuracy calculation is also not clear, what do x,y and function 'I' mean here?

(5) figure 8 is very confusing, the confusion matrix shows there are labels 0 - 6, what do they mean?

In addition, all figures and tables should be cited and described in the paper, e.g., table 4, figure 8. Also, the order is not right, there is no figure 6,7

Author Response

Review contents were well checked.

1. line 254, the authors mentioned 12000 data from pig farms were collected. While it's not clear how the data are applied to the machine learning classification models
->The description was supplemented with 12,080 growth data.

(1) what is the format of these data?
Feeding data in Excel format.
before)
About 12,000 data from pig farms were collected and pre-processed, and in the case of missing data, linear regression method was used to supplement. In order to verify the process of determining the shipping timing for pigs proposed in this paper, 12 models were implemented and 5 models were selected. The accuracy of the model was evalu-ated using the Equation 1 below, and it is possible to plan to improve the accuracy by using additional test data in the future. 
after)
12,040 feeding data from pig farms were collected and pre-processed, and for missing data, the average value was applied, linear regression method was used to supplement. In order to verify the process of determining the shipping timing for pigs proposed in this paper, 5 models were implemented and 2 models were selected. The accuracy of the model was evalu-ated using the Equation 1, and it is possible to plan to improve the accuracy by using additional feeding data in the future.

(2) what is the data distribution? e.g., how many pigs?
12,040 feeding data of farms were used. It has been mentioned in the text.

(3)How does linear regression work on the missing data? It is suggested to give some examples.
Basically, the average value was applied and supplemented through interviews.
Line255, 12,040 feeding data from pig farms were collected and pre-processed, and for missing data, the average value was applied

(4) line 260, the definition of accuracy calculation is also not clear, what do x,y and function 'I' mean here?
(5) figure 8 is very confusing, the confusion matrix shows there are labels 0 - 6, what do they mean?
(4), (5) Considering the concern that confusion may occur, modify it to Test Result and delete the confusion matrix.
before)
A bright number is a high number, and it can be seen that the accuracy is placed at a certain level and can be interpreted as being in a stable range. Accuracy is 792, weighted average is 792. It can be determined that the accuracy of a certain average is maintained. As for precision, the value predicted by the model is in the confidence interval with an average of 0.61. Recall is 0.69, and the reliability of the model is high. The F1-score is the median value of precision and recall, and is 0.65. 
after)
Weighted avg is a high number, and it can be seen that the accuracy is placed at a certain level and can be interpreted as being in a stable range. Accuracy is 792, weighted average is 792. It can be determined that the accuracy of a certain average is maintained. As for precision, the value predicted by the model is in the confidence interval with an average of 0.61. Recall is 0.69, and the reliability of the model is high. The F1-score is the median value of precision and recall, and is 0.65, Test results are shown in Table 5.
Feeding data by pig group should be further supplemented. Then, it is judged that a more accurate confidence interval can be obtained. for balanced data collection, verification of the data collection process and collection results of farms should be strengthened. It is thought that more accurate model learning will be possible if the quantity of data for each group is secured.

In addition, all figures and tables should be cited and described in the paper, e.g., table 4, figure 8. Also, the order is not right, there is no figure 6,7
Figure 8->5, main text Table1->2 modify number

Attached.
thank you
